# Integrative modeling of membrane-associated protein assemblies

Jorge Roel-Touris [1,2], Brian Jiménez-García [1,2✉] & Alexandre M. J. J. Bonvin [1✉]

Membrane proteins are among the most challenging systems to study with experimental structural biology techniques. The increased number of deposited structures of membrane proteins has opened the route to modeling their complexes by methods such as docking. Here, we present an integrative computational protocol for the modeling of membrane-associated protein assemblies. The information encoded by the membrane is represented by artificial beads, which allow targeting of the docking toward the binding-competent regions. It combines efficient, artificial intelligence-based rigid-body docking by LightDock with a flexible final refinement with HADDOCK to remove potential clashes at the interface. We demonstrate the performance of this protocol on eighteen membrane-associated complexes, whose interface lies between the membrane and either the cytosolic or periplasmic regions. In addition, we provide a comparison to another state-of-the-art docking software, ZDOCK. This protocol should shed light on the still dark fraction of the interactome consisting of membrane proteins.

[1] Bijvoet Centre for Biomolecular Research, Faculty of Science-Chemistry, Utrecht University, Utrecht, The Netherlands. [2] These authors contributed equally: Jorge Roel-Touris, Brian Jiménez-García. ✉email: b.jimenezgarcia@uu.nl; a.m.j.j.bonvin@uu.nl

Membrane proteins (MPs) play crucial roles in many biological functions within the cell. Commonly, MPs are classified based on their association mode with biological membranes into two main groups: peripheral membrane proteins that are located on either side of the membrane and are attached to it by non-covalent interactions, and integral membrane proteins (IMPs) that are inserted into the membrane and can be either exposed on only one side of the membrane (monotopic membrane proteins) or span the entire lipid bilayer. The latter, known as transmembrane proteins (TMs), are structurally categorized as α-helical bundles or β-barrels[1]. TMs mostly function as regulators of complex biochemical pathways (receptors and transducers) and/or transporters of molecules (channels and carriers). Only transmembrane proteins can function at both sides of the membrane by forming larger complexes. As such they are not simply passive membrane spanning proteins but play important roles in protein-protein interactions (PPIs), thus making them valuable targets for drug discovery (around 60% of current drug targets are MPs[2]. A well-known example are G-protein coupled receptors (GPCRs), which are involved in many diseases[3]. Those are collected in a specific database (GPCRdb; https://www.gpcrdb.org/)[4].

Over the past years, development of cutting-edge technologies has facilitated the study of previously inaccessible MPs, advancing the field of membrane structural biology. Obtaining high-quality crystals suitable for X-ray crystallography is still far from trivial. Solid-state nuclear magnetic resonance (NMR) spectroscopy, and especially cryo-electron microscopy (cryo-EM), reaching near-atomic resolution, have become central tools to study membrane-associated protein complexes[5,6]. However, experimental conditions such as low-expression profiles and/or high instability outside the native membrane still makes their structural characterization challenging[7]. Despite their large representation in the proteome (in human, nearly a quarter of the genome encodes for MPs[8]), roughly only 1% of all deposited protein structures in the Protein Data Bank[9] (PDB) corresponds to MPs, with 1099 unique protein entries as of July 2020: https://blanco.biomol.uci.edu/mpstruc/. Even less of those have been experimentally solved in complex with their counterpart(s). For all these reasons, membrane protein systems, which are increasingly attracting attention, have been traditionally considered as one of the most difficult type of systems to study by experimental structural biology techniques.

Computational methods offer an attractive alternative for studying membrane systems[10]. Many efforts have been made to develop efficient tools to computationally predict the three-dimensional (3D) atomic structures of membrane-associated proteins and their complexes[11]. Some rely on secondary structure or topology prediction and make use of either knowledge-based statistics or evolutionary information to generate 3D models[12,13]. The simplest computational methods are based on homology modeling. In short, these approaches require a template structure (or multiple) with high sequence similarity to the target sequence, and usually produce very reliable "core" models (corresponding to the TM domains) and less accurate predictions for the extra-cellular loops. Methods such as MEDELLER[14] and Memoir[15] have greatly benefited from the increasing availability of cryo-EM derived structures in the PDB and are inspired on the well-known homology modeling tool MODELLER[16].

Another representative subset of computational methods geared towards modeling complexes are docking-based approaches. Docking commonly includes two different steps, namely sampling and scoring. Sampling is usually referred to as the process of generating (tens of) thousands of possible conformations of a given (bio)molecular complex. This can be done through a number of well-established techniques such as fast fourier transformation (FFT)-based methods included in various docking software, such as GRAMM-X[17,18], ClusPro[19], pyDock[20], and ZDOCK[21]. These methods, however, do not allow for explicit flexibility of the modeled partners due to intrinsic limitations of the FFT sampling. Although this limitation can be partially solved by using ensembles of conformers, it implies higher computational cost. Energy minimization, in HADDOCK[22] and ATTRACT[23] for example, Metropolis Monte-Carlo optimization, e.g., in RosettaDock[24], or artificial intelligence-based algorithms, such as implemented in SwarmDock[25] and LightDock[26], are also used. The sampling process is often followed by a refinement of the docked models for which molecular dynamics- or Monte-Carlo-based protocols are the most commonly used. The generated models are scored with the aim of discriminating between biologically relevant (native) and non-relevant models. This is typically done with a scoring function, which can be based on either physico-chemical properties and/or statistical potentials[27]. Nowadays, with the increasing availability of large pools of docking models such as provided in the CAPRI-DOCK[28] (http://cb.iri.univ-lille1.fr/Users/lensink/Score_set/), PPI4DOCK[29] (http://biodev.cea.fr/interevol/ppi4dock/) and DOCKGROUND[30] (http://dockground.compbio.ku.edu/) databases, machine(deep)-learning scoring functions are gaining interest[31]. Sampling and scoring might be coupled (scoring-driven sampling) or work as independent steps (sampling and then scoring). In the context of membrane protein docking, software such as Rosetta[32], DOCK/PIERR[33], and Memdock[34] include built-in specific protocols to model transmembrane domains using implicit membrane potentials. Besides RosettaMP[35] (for membrane protein design), none of the available membrane-specific computational methods allow for an explicit representation of the lipid bilayer and, therefore, cannot harvest the topological information encoded in it.

In this work, we present an integrative computational approach for modeling membrane-associated protein assemblies (complexes consisting of a membrane-embedded protein and a soluble partner) that combines an efficient, *swarm*-based rigid-body docking by LightDock with a flexible final refinement with HADDOCK to remove potential clashes at the interface. To introduce the topological information provided by the lipid bilayer we make use of an equilibrated coarse-grained membrane into the docking calculations. In that way, we can focus the docking towards binding-competent regions, excluding all irrelevant regions prior to the simulation. This membrane representation has been implemented within the LightDock framework[26]. The sampling in LightDock is based on an artificial intelligence-based *swarm* approach that relies on the metaphor that, in nature, *glowworms* (which represent ligand poses) feel attracted to each other depending on the amount of emitted light (scoring, energetic value of a docking pose). In this way, the docking poses, which constitute the *swarm* of *glowworms* in LightDock, are optimized towards the energetically more favorable ones through the translational, rotational, and anisotropic network model (ANM) spaces. The latter is, however, not available in the membrane docking mode. Sampling and scoring in LightDock are tightly interconnected since the optimization process is score-driven. In its latest official release (version 0.8.1; pypi.org/project/lightdock), LightDock supports the use of information such as mutagenesis and/or bioinformatic predictions to bias the sampling[36]. The LightDock-generated membrane protein models are then refined with HADDOCK via an efficient coarse-grained (CG) protocol[37]. This protocol, originally designed to backmap coarse-grained models to atomistic resolution by morphing atomistic models onto the coarse-grained ones using distance restraints, is very efficient in removing steric clashes while maintaining the original geometry of the docked models.

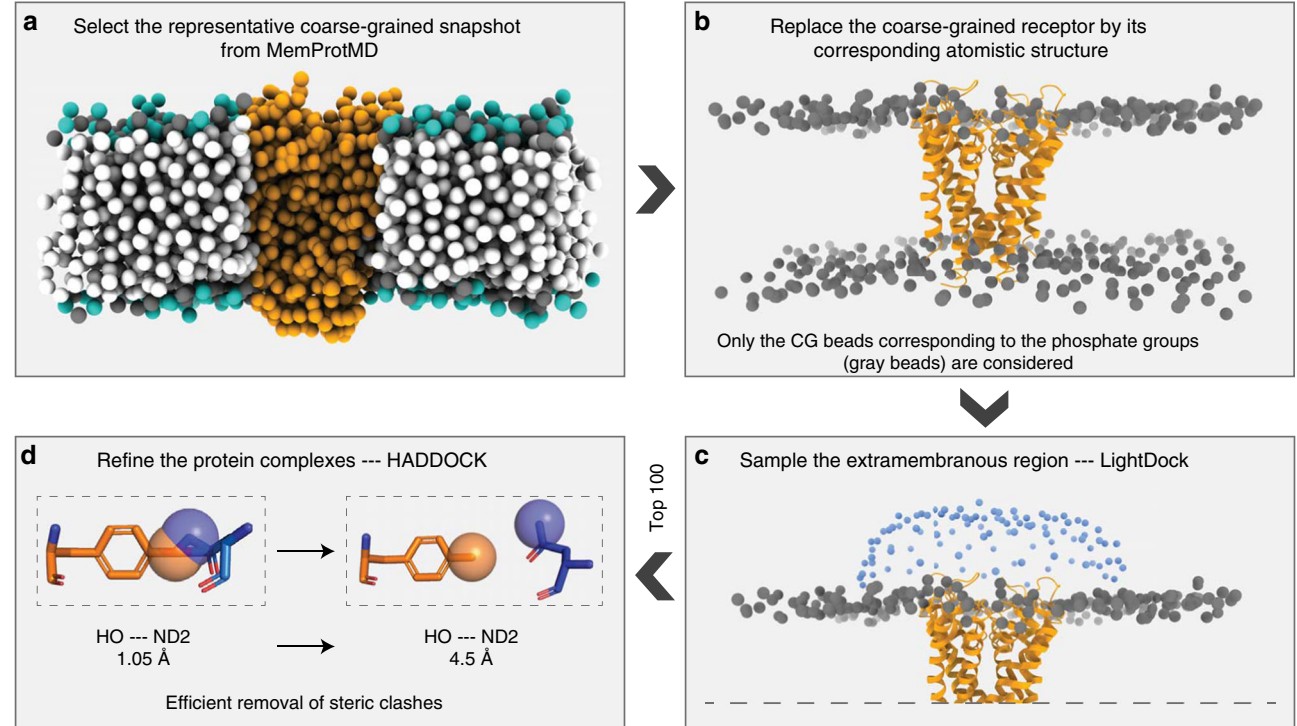

**Fig. 1 Membrane protein integrative modeling workflow. a** The representative coarse-grained membrane snapshot from the MemProtMD database is selected[39] (http://memprotmd.bioch.ox.ac.uk/). **b** The coarse-grained transmembrane receptor is replaced by its corresponding atomistic structure. **c** The binding-competent regions are sampled with LightDock using the membrane defined by beads corresponding to the phosphate positions. The resulting top 100 docked models are selected for final refinement. **d** Refinement with HADDOCK following a coarse-grained to all-atom protocol and final scoring.

We demonstrate the efficiency and performance of this two-step (docking and refinement) membrane-driven protocol on the 18 membrane protein complexes from the MemCplxDB benchmark set[38] (https://github.com/haddocking/MemCplxDB) whose interface lies between the membrane and either the cytosolic or periplasmic regions. We also evaluate how different choices for defining the membrane topology affect the sampling of our protocol, and assess the quality improvement of the generated models after the HADDOCK refinement step. We compare the success rate of this integrative approach and the quality of the generated models with that of another state-of-the-art docking software, ZDOCK[21], for which we test several docking scenarios penalizing ("blocking") regions during sampling and therefore explicitly accounting for the information provided by the membrane. Finally, we discuss the quality of the side chains at the interface of the generated models and propose future developments that could be made for improving the current results.

## Results

**Integrative modeling approach for membrane-associated protein complexes.** We have developed a computational approach for modeling the interaction of membrane-associated protein complexes that accounts for the topological information encoded in the membrane. First, we insert the atomistic transmembrane protein into a pre-equilibrated coarse-grained model of the protein in a lipid bilayer provided by the MemProtMD database[39] (http://memprotmd.bioch.ox.ac.uk/) (see "Methods"; Preprocessing of input structures) and then remove all lipid beads except those representing the phosphate groups. Using this beads layer, we automatically generate a group of independent simulations known as *swarms* over the solvent-exposed receptor surface. Using the capability of the LightDock framework, we thus discard irrelevant sampling regions (see Fig. 1a–c where the geometrical centers of the *swarms* are depicted as blue beads). Next, each

*swarm* is populated with 200 starting random orientations of the soluble ligand (200 is the default number of *glowworms*, the agents of the sampling algorithm). This procedure effectively biases the sampling towards the binding-competent regions on the membrane protein (either cytosolic or periplasmic) and excludes those within the boundaries imposed by the membrane. While Light-Dock can allow for flexibility during docking through normal modes, this option is not supported for membrane-embedded proteins. In the current implementation of the protocol, the membrane-embedded proteins and their beads are considered as a single entity and as such the ANM model is not applicable. For their soluble partners, the inclusion of flexibility is completely functional and might be enabled for the docking process (see "Methods"; Running LightDock in membrane mode). However, in the results hereby presented, the only limited flexibility introduced in the protocol is that of the final refinement using HADDOCK.

For the scoring during the docking simulation with LightDock, we use an adapted version of the DFIRE[40] scoring function that penalizes models penetrating the membrane, specifically those overlapping with any membrane bead (see "Methods"; Implementation of an explicit membrane representation into Light-Dock). We select the top 100 models from the optimization of all *swarms* for a final refinement stage with HADDOCK in order to remove clashes at the interface. This is achieved using an efficient coarse-grained refinement protocol: In short, we first generate the corresponding MARTINI-based[41] coarse-grained representation for each of the docked models to be refined; then, by a combination of energy-minimizations and short molecular dynamics stages, the protocol[37] fits the atomistic structure of each of the components onto the generated CG model of the complex and optimizes the system to remove clashes. This final refinement is performed in the absence of the membrane. The resulting models are then scored and ranked according to the HADDOCK score.

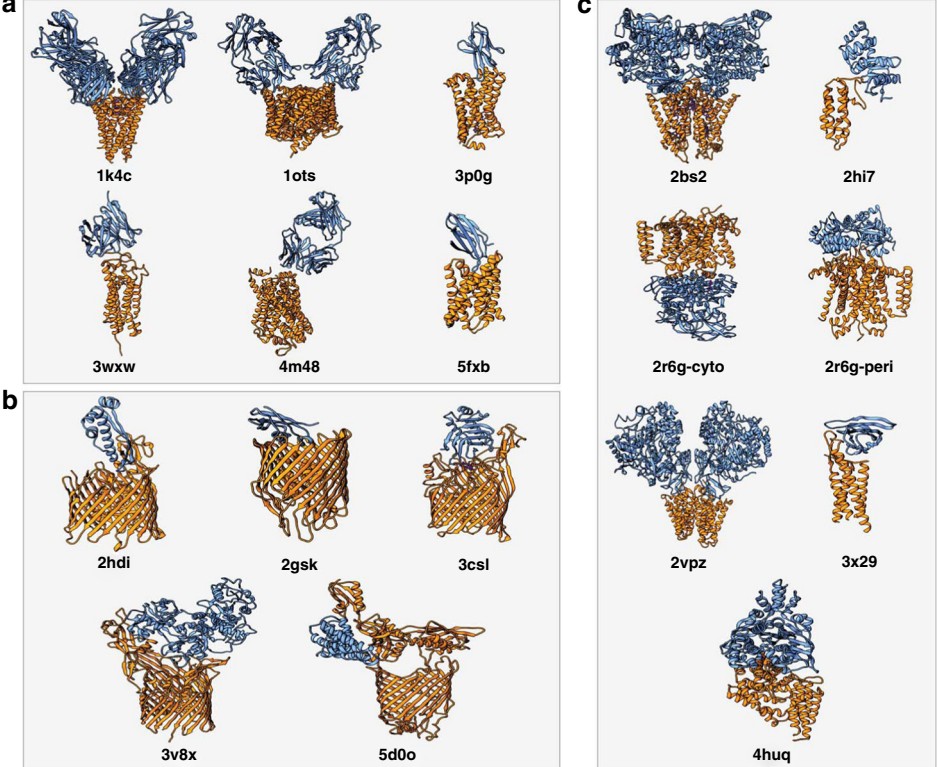

**Fig. 2 View of the 18 transmembrane-soluble protein complexes of the MemCplxDB database[38].** The cases are classified as: **a** Antibodies (the soluble partner is either an antibody or a nanobody), **b** β-Barrel (the transmembrane receptor is a β-sheet barrel) and **c** α-Helical (the transmembrane receptor consists of a bundle of α-helices). The receptors (the membrane proteins) are depicted in orange and the ligands in blue. Unbound initial structures of the 18 cases tested in this work can be found at https://github.com/haddocking/MemCplxDB.

Although in this work our protocol only makes use of the membrane as information source during the modeling, it is fully compatible with the use of a variety of experimental data in the form of residue restraints if this source of information is provided[36].

**Overall performance on the membrane docking dataset.** We have tested the performance of our membrane-driven protocol on the 18 transmembrane-soluble protein complexes of the MemCplxDB benchmark (see "Methods"; Membrane docking dataset and Fig. 2) and compared it with the results of a full sampling in the absence of the topological information provided by the membrane (i.e., Blind docking—see next section). The docking was performed starting from the unbound structures of each constituent, except for *2bs2*, *2vpz*, and *4huq* for which no unbound state structures are available. The success rate was defined as the percentage of cases for which an acceptable or better model was obtained within the top *N* ranked models (see "Methods"; Metrics for the evaluation of model quality and success rate).

For the two most representative top *N* (T5 and T10), our *Membrane* protocol shows an overall success rate of 61.1%, 11 out of 18 complexes, with three cases having medium quality models as shown in Fig. 3. It reaches a maximum of 88.9% for the top 100 predictions. High-quality models are obtained for one α-Helical case within the top 20 (*3x29*) with the best docking pose (ranked at the 12th position) having 70% of the native contacts and 1.0 Å/2.0 Å i-RMSD/l-RMSD (interface/ligand-root mean square deviation) from the reference crystal structure. The highest success rate for either T5 or T10 is achieved for α-Helical complexes. For those, acceptable or higher quality models are generated for

71.4% of the complexes (5 out 7 cases). This performance, however, drops for the β-Barrel category with 40% success rate for T5/10 and 80% for T100 (4 out of 5 cases). Not surprisingly, our protocol fails to deliver near-native models for the case with the largest conformational change (*3v8x*; i-RMSD of 3.42 Å between unbound and bound structures), classified as β-Barrel. For Antibodies (six cases), we used the CDR loops to pre-orient the molecules at the setup step[36], but these were not specifically used for the scoring. For these cases, our protocol generates acceptable and medium quality models for all complexes (100% success rate for T50) with a 33.3% success rate considering the top ranked model (T1) and 66.7% for T5/10.

**The membrane integrative modeling protocol outperforms blind predictions.** For the Blind, membrane-free predictions, LightDock-HADDOCK reaches an overall success rate of 16.7% for the T100 (11.1% of medium quality models), with a moderate performance for T5 and T10 (5.5% and 11.1%, respectively). For one bound case (*4huq*) and one with the second lowest conformational change (*3x29*; 0.67 Å), the Blind protocol does manage to generate models with more than the 30% of the native contacts. For the remaining cases, acceptable models are found only for *2gsk* with the first near-native model ranked at the 39th position with a l-RMSD of 9.47 Å. The top 1 and top 10 performance are similar to what has been reported for HADDOCK using a blind docking scenario[38]. Altogether, the results of the Blind predictions are considerable worse than that of our Membrane protocol in terms of both overall performance and CAPRI-based quality of the generated models. This clearly shows that the use of the membrane topological information to drive the modeling process has a significant impact on the docking performance.

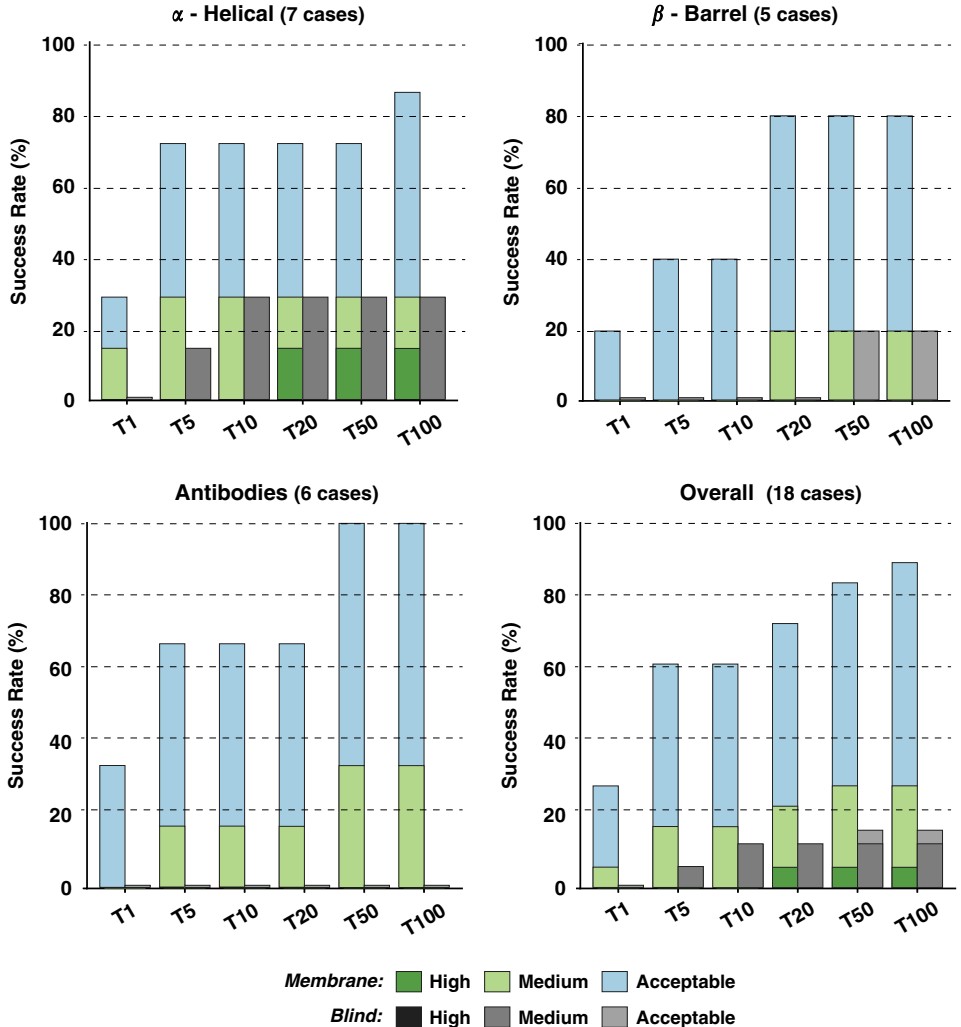

**Fig. 3 Performance of the membrane protein integrative modeling protocol on the 18 cases of the membrane docking benchmark.** Success rates are presented for each of the benchmark categories (α-Helical, β-Barrel, and Antibodies) as well as Overall (including all three different categories). Color coding from blue to green (Membrane) and grayscale (Blind) indicates the model quality (from acceptable to high) as defined based on CAPRI criteria. Source data are provided as an excel file.

**Impact of different membrane definitions on the docking performance.** The results presented so far have been obtained by either defining the membrane based on the phosphate beads positions taken from MemProtMD (Membrane) or by fully blind predictions (i.e., without any membrane). We investigate here how different definitions of the membrane might impact our docking protocol. For that purpose, we have generated two additional artificial bead representations of the membrane based on the average (Average) or minimum (Minimum) z-axis coordinate provided by the equilibrated MemProtMD membrane model. We have compared the docking performance of those different membrane scenarios on the 18 cases from the membrane docking benchmark. As previously, we assess the performance in terms of success rate for each of the selected N tops (see "Methods"; Metrics for the evaluation of model quality and success rate). For the sake of simplicity, we only report the success rate for acceptable or better models.

On average, in the Membrane scenario our simulations have $99 \pm 34$ starting swarms (ranging from 32 to 170), while for Average and Minimum this increases to $134 \pm 28$ and $164 \pm 34$, respectively. This roughly translates into an increase of 7,000 and 13,000 in the number of glowworms (agents of the algorithm representing possible ligand poses) that are handled by the

optimization algorithm during the sampling and scoring processes as compared to the Membrane scenario. As shown in Fig. 4a, for the two most representative tops (T5 and T10) the success rates drop from 61.1% to 44.4% and 27.8% for Average and to 50% and 33.3% for Minimum. This pattern is observed along all selected top N models, which suggests that there is a negative correlation between the number of swarms (and glowworms) and the docking performance. This effect is expected, since the larger the pool of generated poses, the larger the number of possible false positives, which can be selected by the scoring function. For this reason, the optimization of the poses might not always converge towards biological relevant states.

**Penalizing models penetrating the membrane leads to better predictions.** We have also investigated the effect of the scoring penalty on the optimization algorithm during the docking. To do so, we have designed an additional scenario (Filtered), in which the membrane was only used to initially filter the swarms over the receptor surface, but not considered for penalizing models penetrating the membrane (see Fig. 4b). In this case, the success rate of the top 10 is similar to that of Average scenario (50%).

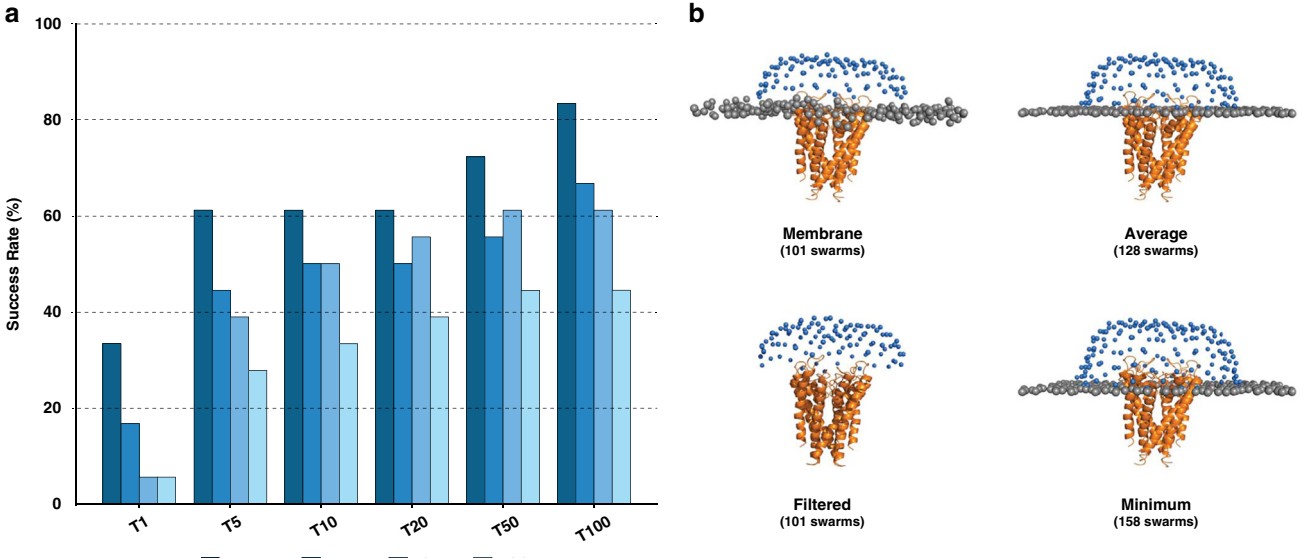

**Fig. 4 Analysis of the impact of different membrane definitions onto the docking performance. a** Bar plot of the performance of the different membrane setups on the 18 cases of the membrane protein docking dataset (i.e., before refinement with HADDOCK). The success rate is defined as the percentage of cases for which an acceptable or higher quality model was found within the selected top $N$. **b** Illustration of the different membrane setups on a representative case of the benchmark (*1k4c*). Source data are provided as an excel file.

However, for higher tops such as T1 and T5, the Filtered scenario performs considerably worse as compared to Membrane (5.5% and 38.8% vs. 33.3% and 61.1%, respectively). This clearly suggests that, while the membrane plays an important role to narrow the conformational search it has also a big impact on the scoring: First, it guides the optimization protocol towards more binding-competent regions and second, it helps identifying near-native states out of the pool of generated docked models.

**The structural quality of the docked models improves after HADDOCK refinement.** We have assessed the quality of the docked models in terms of intermolecular steric clashes. To do so, we have quantified and compared the number of clashes (see "Methods"; Metrics for the determination of steric clashes in a protein complex) present in our docked models before (Light-Dock only) and after refinement with HADDOCK. On average, the top 100 LightDock models have a significant number of clashes ($28.5 \pm 10.0$) compared to those after refinement ($0.6 \pm 0.5$) as shown in Fig. 5a. For some cases, *2bs2*, *2gsk*, *3csl*, and *1ots*, few refined models ranked at positions ≥95 still have a moderate number of those (>25), but these are penalized at the level of the HADDOCK score, which ensures that clashing models will never be ranked at top ranking positions. Overall, this coarse-grained refinement protocol is able to refine and remove more than the 98% of the total number of clashes. As an example, a model before and after refinement is shown in Fig. 5b.

Ideally, a refined complex should not structurally deviate too much from its unrefined counterpart. If this is not the case, the refined interface might significantly differ from the predicted one and therefore loose a relevant predicted state. We have investigated whether our refined models differ from their starting conformations in terms of their interface RMSD of the backbone (i-RMSD). For this, we selected all LightDock models with an i-RSMD ≤ 6 Å from the top 100 predictions for all cases (183 models in total) and compared them to their counterpart after refinement with HADDOCK. As shown in Fig. 5c, the vast majority of points are along the diagonal, which indicates that the backbone of the refined complexes has not significantly moved during the refinement. It is mainly the positions of side chains at

the interface that have been optimized (see Figs. 5b and S2). Points above the diagonal, indicate models that have improved in terms of i-RMSD after refinement. The changes are however limited. Two models (from *2vpz* and *3csl*), however, show a significant improvement of 1.09 Å and 1.85 Å, respectively. In summary, these results show that our coarse-grained refinement protocol is very efficient in removing steric clashes without compromising the quality of the backbone conformation of near-native models.

**Using membrane topological information to drive the docking performs better than post-sampling filtering approaches.** We have also analyzed how our membrane-driven protocol compares to other state-of-the-art docking software. To this end, we selected ZDOCK[21] as docking algorithm of reference for several reasons. First, it is a well-established docking program whose scoring protocol is being trained and continuously tested on a large and relatively heterogeneous benchmark of protein-protein complexes[42]. Second, it allows to mask regions not belonging to the interface. Moreover third, its standalone version (3.0.2) is a fast and easy-to-use tool for systematic benchmarking. Despite that the current version of ZDOCK does not allow to use an explicit representation of the membrane, we have designed three different scenarios in which various levels of information are used to include information about the membrane. In order to mimic our Membrane scenario, we have masked all surface accessible residues below the maximum $z$-coordinate provided by our membrane implementation (ZDOCK-max). Similarly, to compare with our Average and Minimum scenarios, we have masked those residues below the average (ZDOCK-avg) or minimum (ZDOCK-min) $z$-coordinate (Fig. S1). For the Antibodies sub-category, we have also masked all non CDR loops residues. Finally, we run the 18 cases of the membrane docking benchmark in fully blind (default) mode to define the baseline of ZDOCK.

The results for all those scenarios are shown in Fig. S3. The best performance is obtained for the ZDOCK-avg scenario. Comparing this scenario to our Membrane protocol shows that both protocols have an equivalent success rate of 27.7% for the top 1 model, but our protocol clearly performs best for T5/T10 with 61.1% as

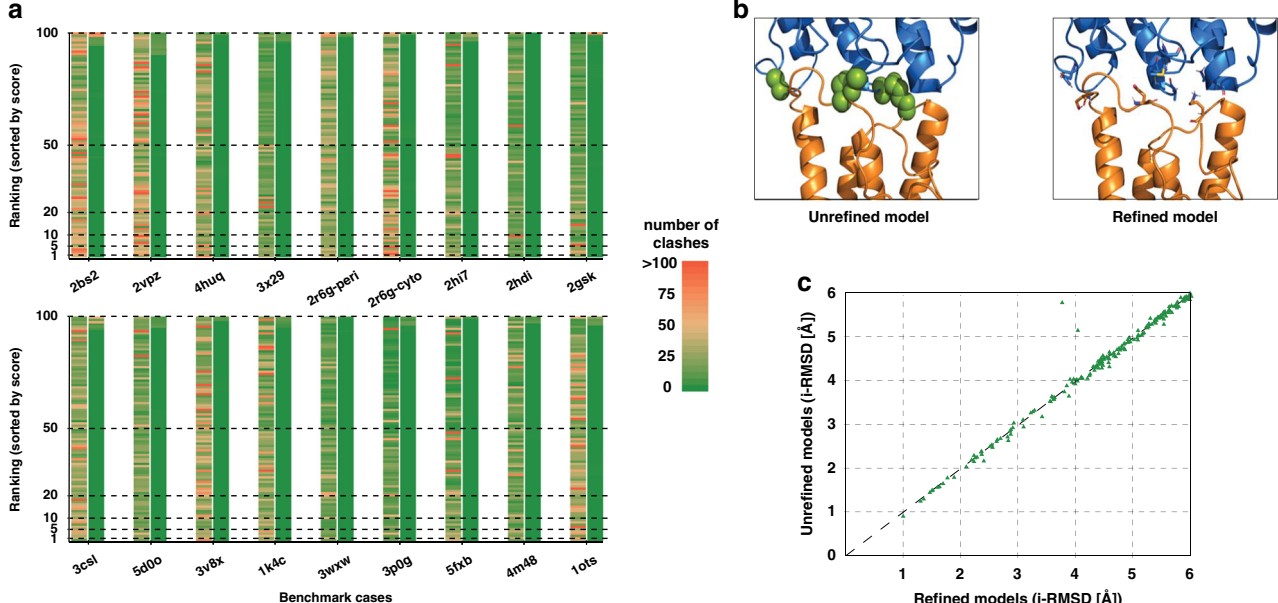

**Fig. 5 Analysis of the quality of the membrane-associated protein models before and after refinement with HADDOCK. a** Stacked bar plot of the top 100 generated models for each of the benchmark cases (18 in total) ranked by their respective score (left–LightDock DFIRE docking score, right-HADDOCK score). For each complex the left bar corresponds to the unrefined models and the right bar to the refined models. The color coding (from green to red) indicates the number of clashes. **b** Illustration of a complex before and after refinement. Green spheres represent atomic clashes. The corresponding side chains are shown as sticks in the refined model. **c** i-RMSD comparison of all models with an i-RMSD ≤ 6 Å before and after refinement (183 in total). Points above the diagonal indicate an improvement in i-RMSD value. Source data are provided as an excel file.

compared to 38.8% for ZDOCK (Fig. 6). Our protocol reaches 88.8% of near-native models for the top 100 compared to the 55.5% for the best performing scenario in ZDOCK. These differences are more remarkable for α-Helical and Antibodies complexes with 71.4% and 66.6% for the top 5 (and top 10), respectively, compared to 28.5% and 33.3% for ZDOCK. In the case of β-Barrel, ZDOCK-avg, however, performs best with 60% in the top 1 (3 out 5) while our protocol starts at 20% for top 1 to reach a maximum of 80% in T20. Based on the well-established CAPRI quality criteria (see "Methods"; Metrics for the evaluation of model quality and success rate), ZDOCK builds high-quality models for the 16.6% of the tested cases (3 out of 18, while only one high-quality model is obtained in our case) with 2 of them ranked within the top 10 predictions. These two cases correspond to the β-Barrel complex with the smallest conformational change (2hdi; i-RSMD of 0.361 Å between unbound and bound structures) and a α-Helical bound case (4huq).

## Discussion

In this work, we have developed and tested an integrative modeling protocol to build membrane-associated protein assemblies. The protocol, which specifically accounts for the topological information encoded in the membrane, combines the capability of the LightDock framework to discard non-binding regions prior to the docking, with an efficient coarse-grained refinement via HADDOCK to remove clashes. As previously demonstrated, including information during docking not only outperforms the scenario where data are only used to discard models (post-simulation approaches), but also reduces significantly the computational cost[36], in this particular case by an average factor of 75% over the 18 complexes considered. Our membrane-driven protocol shows a much better performance in generating native-like structures for the vast majority of the tested cases than when the membrane is neglected (Blind). It achieved this by both filtering the initial swarm configurations and including a membrane penalty term into the scoring, which helps both the optimization

algorithm and the scoring of the docked models. Altogether, our findings reinforce the well-accepted notion that the integration of (experimental) data, in this case membrane topological data, into the docking calculations improves the performance of modeling approaches.

We have also investigated how different ways of defining the membrane topological information across the z-axis affect the sampling of the conformational space. Our protocol performs best when an equilibrated and simulated bilayer is incorporated into the sampling. This limits the number of swarms (and therefore glowworms), which, in turns, allows the optimization algorithm to identify more biological relevant states compared to less restricted scenarios such as Average and Minimum. This behavior is explained by the fact that in LightDock, sampling and scoring are closely interconnected since the optimization of the ligand poses (in rotational and translational spaces) is driven towards better scoring conformations. In other words, the reduction of potential false positives leads to an increase in the performance of the search algorithm.

We have analyzed the quality improvement of our docking predictions after refinement using a simple definition of steric clashes. We have shown that our refinement protocol leads to the removal of (almost) all clashes while keeping the backbone conformation almost unaltered, with no more than 0.25 Å i-RMSD for the most altered conformations (rare cases). As a consequence of the refinement, the side chains might suffer from bad conformations introduced by the removal of clashes and move away from the native conformation in the complex. To check this, we have analyzed the impact of the coarse-grained refinement on the side-chain i-RMSD and the fraction of native intermolecular contacts they form. As shown in Fig. S2, the refined models do not significantly loose native intermolecular contacts as estimated by the Fnat metric and their side-chain i-RMSDs even slightly improve.

Some knowledge of the putative binding interface is known to help the modeling of biomolecular interactions, often allowing to

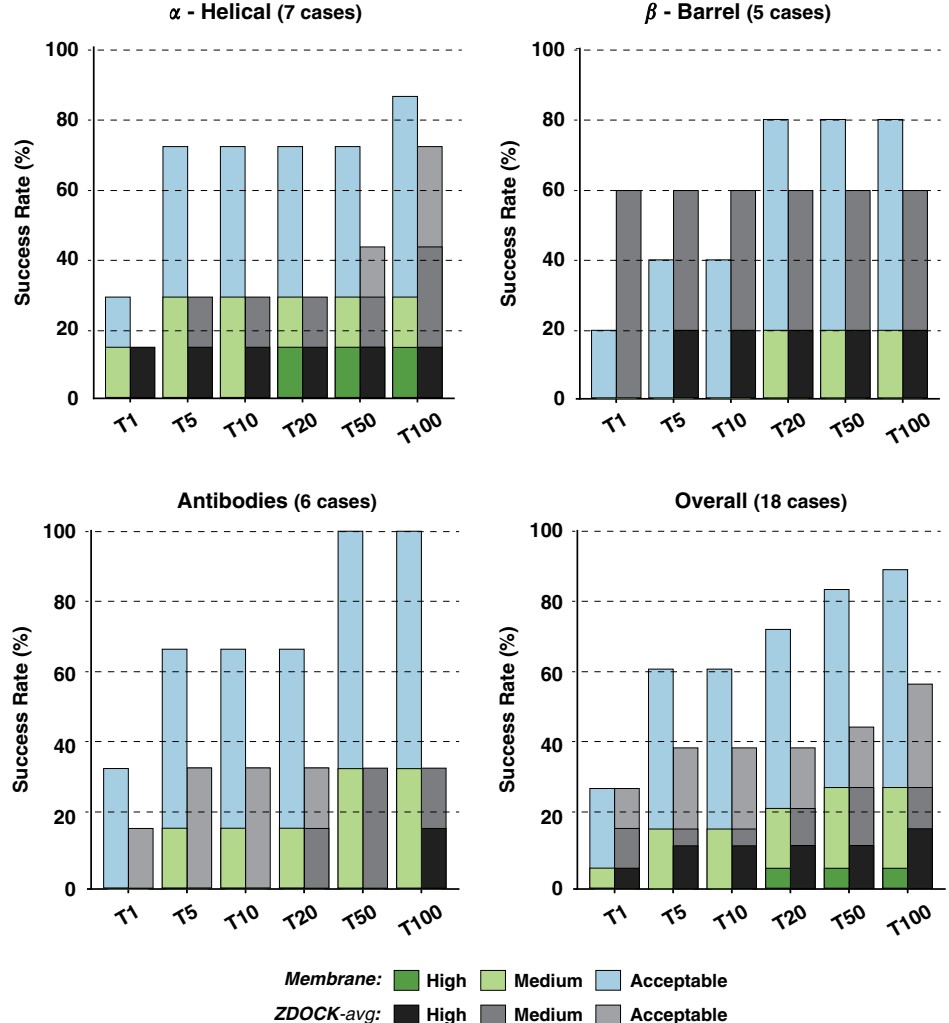

**Fig. 6 Performance of the membrane integrative modeling protocol compared to the best ZDOCK scenario.** Success rates are presented for each of the benchmark categories (α-Helical, β-Barrel and Antibodies) as well as overall (including all three different categories) for the membrane protocol (Membrane, color coding from blue to green) and the best ZDOCK scenario (ZDOCK-avg, grayscale) on the 18 cases of the membrane docking benchmark. The color coding/grayscale indicates the model quality (from acceptable to high) according to CAPRI criteria. Source data are provided as an excel file.

generate more accurate models. This information can come from a variety of experimental of bioinformatic data, such as, for example, NMR chemical shift perturbations, mutagenesis data, H/ D exchange and cross-linking data obtained by mass spectroscopy or sequence conservation, among others[43]. Besides of the topological information encoded in the membrane, our protocol can also incorporate information about interfaces. As an example, we assumed that three interacting residues of the soluble partner are known and defined those in LightDock (see Table S1 in Supplementary Information). The results, shown in Fig. S4, show that this does not significantly change the performance as the T100 remains constant (75% success rate) with a slight improvement in the top 10 predictions (75% as compared to 66.6%) and a slight decay in T5 and T1 (25% and 58.3% as compared to 33.3% and 66.6% for the Membrane-rst and Membrane categories, respectively). It is worth noting that these variations are not that significant as they are caused by only a single case difference because of the limited size of the benchmark.

In this work, we have only focused on the modeling of membrane-associated protein assemblies. In cellular environments, however, some soluble proteins might associate with membranes in order to stabilize and/or carry out their function. These types of interactions have only been studied in a handful of systems such as signaling factors or nuclear receptors, due to the lack of more generic approaches that can be used to characterize a broader range of lipid-protein interactions[44]. Our work could be extended to build realistic models of membrane-associated protein complexes. This would require extra effort to develop a scoring function that accounts for protein–lipid interactions. Such membrane-specific scoring functions have been already shown appropriate for membrane protein structure prediction and design purposes[45] and might also represent a significant advance for membrane-associated protein docking protocols. Looking ahead, a larger benchmark set will enable broader energy function development and optimization, which should eventually cover protein–lipid interactions too. In terms of software integration, LightDock, as a sampling algorithm, could be included within the future modular version of the HADDOCK software and eventually offer an alternative to its default rigid-body sampling step. This would further extend HADDOCK modeling capabilities to account for the use of membrane-based bilayers. Note that HADDOCK has already been used with explicit membranes (nanodisks or micelles) to study the binding and orientation of proteins onto the lipid surface[46–48]. These are however isolated cases and no systematic testing as performed here has yet been done.

In summary, we have developed an integrative modeling protocol for membrane-associated protein assemblies that accounts for the topological information provided by the membrane in the modeling process. It makes use of a membrane-derived bead bilayer during the sampling step with LightDock. Clashes resulting from the rigid-body docking are successfully removed by refinement with HADDOCK while preserving the quality of both backbone and side chains conformations at the interface. Importantly, while the present protocol only makes uses of the membrane to drive the modeling, it is fully compatible with the use of other sources of information such as mutagenesis and/or bioinformatic predictions in the form of residue restraints to further guide the docking.

## Methods

**Membrane docking dataset**. We selected all complexes from the MemCplxDB database[38] (https://github.com/haddocking/MemCplxDB) whose interface lies between the membrane and either cytosolic or periplasmic regions. This selection yielded a dataset of 18 cases (See Fig. 2), which were further classified into:

- α-Helical: complexes whose receptor assembles as a α-helical bundle.
- β-Barrel: complexes whose receptor forms an antiparallel β-sheet composed tandem of repeats.
- Antibodies: complexes whose soluble ligand is an antibody or nanobody.

**Preprocessing of input structures**. We make use of an equilibrated coarse-grained representation of the membrane to include topological information in our modeling procedure. For this, for each benchmark case, we obtain a representative coarse-grained snapshot of the transmembrane protein inserted into a simulated lipid bilayer (MARTINI representation[41]) from the MemProtMD database (Fig. 1a)[39] (http://memprotmd.bioch.ox.ac.uk/). For the sake of simplicity and for saving computational resources, we remove all lipid beads except those representing the phosphate groups, which, to some extent, represent the most external layers. Then, we replace the coarse-grained TM receptors by their corresponding atomistic structure (Fig. 1b). When needed, we remove beads overlapping or clashing (< 2.5 Å distance) with any heavy atom of the transmembrane protein once inserted into the membrane (*1ots*, *2gsk*, *2hi7*, *4m48*, and *3wxw*).

**Implementation of a coarse-grained membrane in LightDock**. To allow for the use of a coarse-grained membrane within the LightDock framework, we added new logic for the two different stages namely: The internal preparation of the molecules (at the *setup* level) and the actual simulation (at the scoring level). In the first stage, *setup*, we have added a new flag (-*membrane*) to activate the filtering of initial swarms (independent centers of simulation) according to the topological information of the membrane (no swarms will be generated below it). The protocol will detect the number of bead membrane layers provided by the user and select the upper one. For that purpose, it is expected that the user will provide the structure in PDB format and by a cenital plane point of view (the *z*-axis is perpendicular to the membrane plane, that is the default view when saved by PyMol[49] for example). In case the lower layer is the target of interest, the system should be rotated by 180° around the *x*- or *y*-axis. During the simulation, we have included a term into the scoring scheme so that docking models in which the ligand penetrates the membrane are penalized and will be forced to optimize towards more favorable poses. In our case, we have defined a very unfavorable potential value for the membrane beads in the DFIRE scoring function used by LightDock (−999.0 - the more negative the value is, the worse becomes the score), in order to penalize models whose ligand's position is incompatible with the provided membrane model.

**Running LightDock in membrane mode**. LightDock execution consists of two steps: *setup* and *simulation*. In the first step, *setup*, the user provides to the *light-dock3_setup.py* command line tool the receptor and ligand structures in PDB file format, together with the number of *swarms*, glowworms per *swarm* and other options such as removing hydrogen atoms and/or enabling ANM. In this new version of LightDock, a -*membrane* flag has been implemented in order to filter out swarms not compatible with the simulated coarse-grained membrane. For each of the filtered swarms, if residue restraints information is provided (as it is the case for the CDR loops for antibody-antigen complexes), this is used for pre-orienting the ligand poses as previously described[36]. In this work, the number of initial swarms used is 400 (default - many of them will be filtered by the membrane protocol) and the number of glowworms 200 (default). Hydrogen atoms are also removed as they are not supported by the DFIRE scoring function. Although not used in this work, the flexibility provided by the ANM implementation in LightDock is supported for the ligand (soluble) molecules (not the membrane-embedded proteins as those are considered as one entity together with the beads). This can be activated as: -*anm* -*anm_rec = 0* -*anm_lig = X*, where X indicates the number of non-trivial normal modes to be considered (being 10 the recommended value). When the setup step finishes, the docking simulation is ready to be started. A second command line tool,

*lightdock3.py*, performs the simulation for the number of steps provided by the user (100 in this work) using the DFIRE scoring function and running in parallel depending on the number of cores specified. Once the simulation finishes successfully, predicted poses are generated (*lgd_generate_conformations.py*) and clustered (*lgd_cluster_bsas.py*) according to the default LightDock protocol. Finally, the *lgd_rank.py* command line tool generates a ranking of the top clustered predictions according to LightDock. An exhaustive tutorial of the different steps of the protocol can be accessed online at https://lightdock.org/tutorials/membrane.

**LightDock computational time requirements**. The average run time of a Light-Dock simulation for the benchmark set is 197 min with minimum and maximum values of 22 and 427 min, respectively (as measured using 48 AMD Opteron 6320 2.8 GHz CPU cores). These times are for the current Python version of LightDock. A new port of the code to the Rust programming language (https://github.com/lightdock/lightdock-rust) shows a general speedup of 8 to 10 times compared to the Python version, which should make it possible to provide it as a web-based server in a near future.

**Coarse-grained refinement in HADDOCK**. For the local installation, models must be converted into their coarse-grained representation. This is done via an *in-home* script included in the *CGtools* directory of the HADDOCK2.4 distribution as: "*python aa2cg.py model.pdb*". As output, the script generates the MARTINI-based CG model (*model_cg.pdb*) as well as a restraints file in the form of *model_cg_to_aa.tbl*, which includes the mapping of the generated coarse-grained beads to their corresponding atoms. The *atom-to-bead* restraints files of the different CG models must be combined into a single file (e.g., *cg-to-aa.tbl*) that will be used by HADDOCK to restore the atomistic resolution. In order to perform the refinement, a handful of parameters within the HADDOCK parameter file (*run.cns*) must be adapted as follows assuming that 100 models will be refined:

- *rotate180_it0 = false* (to skip sampling 180° complementary interfaces)
- *crossdock = false* (to refine receptor – ligand from the structures provided)
- *rigidmini = false* (to skip *it0* stage)
- *randorien = false* (to skip *it0* stage)
- *rigidtrans = false* (to skip *it0* stage)
- *ntrials = 1* (to skip *it0* stage)
- *structures_0 = 100* (for *it0* stage)
- *structures_1 = 100* (for *it1* stage; must always be ≤ than *structures_0*)
- *anastruc_1 = 100* (for analysis purposes at *it1* stage)
- *waterrefine = 100* (for *itw* stage; this is the number of final output models)
- *initiosteps = 0* (to skip *it1* stage)
- *cool1_steps = 0* (to skip *it1* stage)
- *cool2_steps = 0* (to skip *it1* stage)
- *cool3_steps = 0* (to skip *it1* stage)
- *dielec_0 = cdie* (to switch a constant dieletric constant when CG is used)
- *dielec_1 = cdie* (to switch a constant dieletric constant when CG is used)

For setting up the refinement on the HADDOCK2.4 webserver version, a tutorial can be found at http://www.bonvinlab.org/software/haddock2.4/tips/advanced_refinement/. Note that on the server coarse-graining should be enabled under the *Input data* tab.

The refined models are scored and ranked according to the default HADDOCK score, which is a linear weighted combination of terms as:

$$\text{HADDOCK}_{\text{score}} = 1.0^* E_{\text{vdw}} + 0.2^* E_{\text{elec}} + 0.1^* E_{\text{AIR}} + 1.0^* E_{\text{desolv}} \qquad (1)$$

where $E_{\text{vdw}}$ and $E_{\text{elec}}$ are the van der Waals and electrostatic energies terms calculated using a 12-6 Lennard-Jones and Coulomb potential respectively, with Optimized Potentials for Liquid Simulations (OPLS) nonbonded parameters, $E_{\text{AIR}}$ is the ambiguous interaction restraints energy, $E_{\text{desolv}}$ is an empirical desolvation score[50]. Note that in this protocol, since we are only refining the model and not providing any restraints, the $E_{\text{AIR}}$ term is not contributing to the final score. An example of the HADDOCK parameter files to run the refinement (*run.param* and *run.cns*) can be found at: https://github.com/lightdock/membrane_docking/tree/master/refinement/example.

**Metrics for the evaluation of model quality and success rate**. The quality of the models is assessed according to the well-accepted CAPRI criteria[51]. Docking models are classified as high (***), medium (**), or low (*) quality according to their similarities with the native structure by calculating the interface and ligand root mean-square deviations (i-RMSD and l-RMSD as calculated with ProFit version 3.1: http://www.bioinf.org.uk/software/profit/) and the fraction of native contacts (Fnat) as:

- High: Fnat ≥ 0.5 and i-RMSD ≤ 1 Å or l-RMSD ≤ 1 Å,
- Medium: Fnat ≥ 0.3 and 1 Å < i-RMSD ≤ 2 Å or 1 Å < l-RMSD ≤ 5 Å,
- Acceptable: Fnat ≥ 0.1 and 2 Å < i-RMSD ≤ 4 Å or 5 Å < l-RMSD ≤ 10 Å and
- Incorrect: Fnat < 0.1 or i-RMSD > 6 Å or l-RMSD > 10 Å.

The overall success rate is defined as the percentage of benchmark cases with at least one acceptable or better model within a given Top N (N = 1, 5, 10, 20, 50 100).

**Metrics for the determination of steric clashes**. We define a steric clash as any heavy atom-heavy atom intermolecular contact shorter than 2.5 Å (i.e., hydrogens excluded). Using this definition of clashes, we sought to investigate whether our coarse-grained refinement protocol in HADDOCK leads to higher quality structures (i.e., less absolute number of clashes) as compared to those generated from the docking step with LightDock. To do so, for each of the benchmarked cases we quantified and compared, on a per model basis, the number of clashes present on the top 100 docked models before and after refinement.

**Reporting summary**. Further information on research design is available in the Nature Research Reporting Summary linked to this article.

## Data availability

A reporting summary for this Article is available as a Supplementary Information file. Code to reproduce results presented in this manuscript as well as the membrane-associated protein docked/refined models can be found at https://github.com/lightdock/membrane_docking[52]. The unbound structures from the MemCplxDB benchmark set tested in this manuscript can be found at https://github.com/haddocking/MemCplxDB[53]. An online tutorial concerning structure preparation and docking with LightDock is available at https://lightdock.org/tutorials/membrane. Further reference and help on how to refine models with the new Haddock2.4 server can be found at http://www.bonvinlab.org/software/haddock2.4/tips/advanced_refinement/. Source data are provided with this paper.

## Code availability

The full HADDOCK version 2.4 is freely accessible as a webserver at: https://wenmr.science.uu.nl/haddock2.4 and as a local installation (See bonvinlab.org/software/haddock2.4). LightDock is open source software released under GPLv3 license, available at: https://github.com/lightdock/lightdock/. LightDock is also available at the Python Package Index Repository (PyPI) https://pypi.org/project/lightdock/ and can be easily installed from the command line using *pip*.

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

## Acknowledgements

We would like to thank all the Computational Structural Biology group at Utrecht University and specially Dr. Panagiotis Koukos for all the useful insights and fruitful discussions. This work has been done with the financial support of the Dutch Foundation for Scientific Research (NWO) (TOP-PUNT grant 718.015.001) and the European Union Horizon 2020 projects BioExcel (675728, 823830) and EOSC-hub (777536).

## Author contributions

J.R.T. has designed the protocol, performed the experiments, analyzed the data, and co-written this manuscript. B.J.G. has co-designed the protocol, developed the new features required for the LightDock software and co-written this manuscript. A.M.J.J.B. has co-designed the protocol, developed the HADDOCK required modules, supervised and directed the experiments, and co-written this manuscript.

## Competing interests

The authors declare no competing interests.
