## [Peer Review File · Nature Communications]

REVIEWER COMMENTS

Reviewer #1 (Remarks to the Author):

Roel-Touris et al. present a paper that discusses the application of LightDock and HADDOCK to propose membrane protein complexes.

This is an excellent paper that is well written with well prepared figures. It also details a very important tool that will be extremely useful for the membrane protein community.

It would be interesting to know whether additional restraints eg bioinformatic or experimental-derived contacts could further enhance the protocol. There are a couple of sentences that do mention this but an example of this would be of interest for comparison - ie would this enhance the overall methodology or impede it?

How does the methodology work for membrane protein contacts where part of the soluble domain snorkels into the membrane to form contacts with the TM helices and not just the solvent exposed loops?

Clearly the methodology has a greater likelihood of success the more results that are generated. It's not clear how one would therefore use this if the solution is unknown? Could this be discussed in more detail - sorry if I missed this in the current version of the text.

You note that docking to membranes could also be possible. Could you also consider soluble docking to membrane proteins with either a ligand or lipid bound that enhances binding?

Minor comments:

1. In places MemProtDB should be changed to MemProtMD and the reference used should be Newport 2019 NAR.
2. The beta symbol used looks odd throughout the manuscript.

Thanks for developing this excellent tool.

Reviewer #2 (Remarks to the Author):

The manuscript deals with the modeling of membrane protein complexes. It describes a docking algorithm for a pair of proteins, where one of them (receptor) is a transmembrane protein and the other (ligand) is outside of the membrane.

The presented algorithm is a combination of two methods, developed previously by the co-authors. These are the LightDock swarm-based docking algorithm, which is used for the initial sampling of the candidate docked ligand poses, and the HADDOCK method, which is used here for scoring and ranking of the resulting hypotheses.

First they insert the transmembrane receptor into a coarse grained model of the protein in the lipid bilayer and then remove the lipid beads except for the phosphate ones. This procedure restricts the sampled ligand poses to be positioned outside of the membrane.

To summarize, the algorithm includes 3 major steps : (i) restriction of the ligand sampling poses; (ii) rigid docking by the LightDock docking scheme, which internal scoring function has been adapted to penalize membrane penetration, in order to generate docking hypotheses; (iii) scoring and ranking of the obtained docking hypotheses by the earlier developed HADDOCK scoring method.

The authors have applied their method in 18 cases taken from the MemCplx benchmark and the quality of the results was evaluated according to the CAPRI Docking Challenge criteria. The results have been compared with the ZDOCK docking scheme on the same examples, restricting the ligand

docking poses to be out of the membrane as well. It should be noted, though, that the ZDOCK scheme was not otherwise geared or optimized towards membrane protein docking. As expected the new scheme outperformed ZDOCK.

The manuscript provides a useful incremental advance of the state of the art.

Remarks :

1. It is desirable to establish a user friendly web-server in order to facilitate the application of the method by biological users. Currently a set of downloadable standalone programs is presented. Is there a problem to establish a web server due to execution time constraints?
2. The text does not mention efficiency of the computational procedures and their CPU runtimes. It is important to supply this information.
3. It is mentioned in the text that the limited normal mode analysis based conformational flexibility that is provided in LightDock, is not applied in the membrane docking case. Why is it so? In the original LightDock paper this flexibility was presented as an advantage.

Dear editor,

We would like to thank the reviewers for their very positive comments and suggestions to improve our manuscript. Below, we provide detailed responses to the various comments.

Reviewer #1 (Remarks to the Author):

Roel-Touris et al. present a paper that discusses the application of LightDock and HADDOCK to propose membrane protein complexes.

This is an excellent paper that is well written with well prepared figures. It also details a very import tool that will be extremely useful for the membrane protein community.

It would be interesting to know whether additional restraints eg bioinformatic or experimental-derived contacts could further enhance the protocol. There are a couple of sentences that do mention this but an example of this would be of interest for comparison - ie would this enhance the overall methodology or impede it?

The membrane protocol is compatible with the use of extra information in the form of residue restraints as previously described (<https://doi.org/10.1093/bioinformatics/btz642>). For the *Antibodies* category we have already made use of such information in the form of the residues belonging to the HV-loops. As a result, our protocol generates near-native models for the totality of the tested cases (6 complexes). However, for the remaining cases no extra information was used.

Following the reviewer's suggestion, we have revised our manuscript adding a paragraph in the discussion section and a new figure (Supplementary Information Fig. S4 and associated Table S1) in which we have analyzed how the use of the experimental-like information impacts the performance of the docking process in the remaining 12 complexes (i.e. excluding *Antibodies*). To this end, from the set of known contacts between receptor and ligand as calculated with a 3.9 Å distance cutoff, we have selected 3 residues on the soluble protein to be used as "active restraints" in LightDock (those are listed in Table S1).

As previously described, these residues are used to first, pre-orient the initial ligand poses (*glowworms*) in the *setup* stage and then, to bias the scoring according to the percentage of satisfied restraints during the optimization step. We analyzed the new set of simulations, referred to in the text as *Membrane-rst*, according to the CAPRI criteria and we provide the results compared to our membrane protocol (*Membrane*) in Fig. S4 in Supplementary Information. The impact of adding such restraints is rather limited in this case, but this might be related to the limited size of our benchmark.

How does the methodology work for membrane protein contacts where part of the soluble domain snorkels into the membrane to form contacts with the TM helices and not just the solvent exposed loops?

In the current manuscript, we have focused on the modeling of membrane-associated assemblies, consisting of a membrane-embedded protein and a soluble partner, the latter mainly contacting solvent-exposed loops of the membrane receptor. The implemented membrane penalty term helps the optimization algorithm during the docking process by penalizing models penetrating in a high degree

the membrane. The reviewer correctly points out that there might be cases where not only the solvent exposed TM loops make contacts but the TM helices too. The current implementation does not allow to model such cases as the penalties will be high. A possible solution to this could be to define specific regions of the soluble protein which are not allowed to penetrate the membrane. This however does require *a priori* knowledge about those regions. This is an interesting idea which we might pursue in the future, provided we can extend the benchmark to have enough representative cases.

Clearly the methodology has a greater likelihood of success the more results that are generated. It's not clear how one would therefore use this if the solution is unknown? Could this be discussed in more detail - sorry if I missed this in the current version of the text.

As any modeling method, there is no guarantee of success. This is also clear from the reported success rates as a function of the number of considered models, which is why it will remain important to consider more than one model. As we often advertise in workshops and training events, this kind of modeling should not be seen as the end of the road, but as a way of generating testable hypothesis by further experiments.

You note that docking to membranes could also be possible. Could you also consider soluble docking to membrane proteins with either a ligand or lipid bound that enhances binding?

We assume here that the reviewer describes the case where a ligand or bound lipid is found in between the two proteins, enhancing the binding. If such a molecule is already present in the unbound forms of the protein it could be incorporated during the docking provided the used scoring function can account for it, which is currently not the case. If the binding site is not known, and the ligand should be docked simultaneously with the two proteins, this would consist of a three-body docking, which is not supported in LightDock. So, at this time, such a scenario is not supported.

Minor comments:

1. In places MemProtDB should be changed to MemProtMD and the reference used should be Newport 2019 NAR.

We have replaced MemProtDB by MemProtMD and changed its reference (#39) accordingly.

2. The beta symbol used looks odd throughout the manuscript.

We have replaced the symbol for better readability.

Thanks for developing this excellent tool.

We would like to thank again the reviewer for the constructive comments and discussions.

Reviewer #2 (Remarks to the Author):

The manuscript deals with the modeling of membrane protein complexes. It describes a docking algorithm for a pair of proteins, where one of them (receptor) is a transmembrane protein and the other (ligand) is outside of the membrane.

The presented algorithm is a combination of two methods, developed previously by the co-authors. These are the LightDock swarm-based docking algorithm, which is used for the initial sampling of the candidate docked ligand poses, and the HADDOCK method, which is used here for scoring and ranking of the resulting hypotheses.

First they insert the transmembrane receptor into a coarse grained model of the protein in the lipid bilayer and then remove the lipid beads except for the phosphate ones. This procedure restricts the sampled ligand poses to be positioned outside of the membrane.

To summarize, the algorithm includes 3 major steps : (i) restriction of the ligand sampling poses; (ii) rigid docking by the LightDock docking scheme, which internal scoring function has been adapted to penalize membrane penetration, in order to generate docking hypotheses; (iii) scoring and ranking of the obtained docking hypotheses by the earlier developed HADDOCK scoring method.

The authors have applied their method in 18 cases taken from the MemCpIX benchmark and the quality of the results was evaluated according to the CAPRI Docking Challenge criteria. The results have been compared with the ZDOCK docking scheme on the same examples, restricting the ligand docking poses to be out of the membrane as well. It should be noted, though, that the ZDOCK scheme was not otherwise geared or optimized towards membrane protein docking. As expected the new scheme outperformed ZDOCK.

The manuscript provides a useful incremental advance of the state of the art.

Remarks:

1. It is desirable to establish a user friendly web-server in order to facilitate the application of the method by biological users. Currently a set of downloadable standalone programs is presented. Is there a problem to establish a web server due to execution time constraints?

We agree with the reviewer that a user-friendly web-server will facilitate the application of the membrane protocol by non-expert users. We have a long history of providing such services in our group. Providing production-level, high quality services is however not trivial, but clearly something we will do in the future. We are actually already working on a web-based application for the LightDock software, but this falls outside the scope of this manuscript. In particular, the re-write of LightDock into the Rust programming language gave us a two order of magnitude performance in terms of memory usage and a speedup of 8 to 10 times (a timing section has been added to the Methods section). With that we are confident we can offer in a not too far future a full web-based implementation of our protocol, including the connection to the HADDOCK server for the refinement part.

2. The text does not mention efficiency of the computational procedures and their CPU runtimes. It is important to supply this information.

While some information was already provided in section 3.1 of the full demo for complex 3x29 (https://github.com/lightdock/membrane_docking/blob/master/demo/README.md), we agree with the reviewer that computational efficiency is important for the final users. We have therefore added run time information in the Methods section under the subsection "LightDock computational time requirements".

3. It is mentioned in the text that the limited normal mode analysis based conformational flexibility that is provided in LightDock, is not applied in the membrane docking case. Why is it so? In the original LightDock paper this flexibility was presented as an advantage.

In LightDock, molecular flexibility in the backbone (to a certain degree) is introduced using the Anisotropic Network Model as calculated by the Python library ProDy (<http://prody.csb.pitt.edu/>). In the current version of the membrane protocol, the transmembrane receptor and the membrane beads are considered as the same entity and as such, ANM would not work since the membrane beads are not recognized by ProDy. For this reason, we have not used ANM in this work and performed the docking in rigid-body mode. For a future version of LightDock we will consider decoupling the membrane beads from the transmembrane proteins to allow for ANM also on the membrane-embedded protein.

Note that the ANM model can be used for the soluble component. As a test, we have run three cases (one per category) with varying degree of conformational change between unbound and bound forms, allowing flexibility on the soluble partner (ligand). We provide a comparison of the results (number of acceptable or better models) with and without ANM in the following tables:

Rigid-body

Case	Conformational change (Å)	T1	T5	T10	T20	T50	T100
4huq	0.0	1	1	1	1	1	1
2gsk	0.867	0	1	2	2	2	2
4m48	2.335	0	0	0	0	0	5

ANM in soluble protein (ligand)

Case	Conformational change (Å)	T1	T5	T10	T20	T50	T100
4huq	0.0	1	3	3	3	3	3
2gsk	0.867	0	1	1	2	2	2
4m48	2.335	0	0	0	0	0	5

In terms of success rate, there are no striking differences between the “rigid-body” and the “ANM in soluble protein (ligand)” scenarios for these three particular cases, but we do observe a few more acceptable models when ANM is used. From this small study, we cannot conclude whether the inclusion of flexibility through NMA constitutes a clear advantage for the modeling of membrane-associated protein assemblies but it demonstrates is fully compatible and functional.

We have clarified this point in the revised manuscript adding the following statement, under the *Integrative modeling approach for membrane-associated protein complexes* section:

“In the current implementation of the protocol, the membrane-embedded proteins and their beads are considered as a single entity and as such the ANM model is not applicable. For their soluble partners, the inclusion of flexibility is completely functional and might be enabled for the docking process (See Material and Methods; Running LightDock in membrane mode). However, in the results hereby presented, the only limited flexibility introduced in the protocol is that of the final refinement using HADDOCK.”

and in the Materials and Methods section: *Running LightDock in membrane mode:*

“Although not used in this work, the flexibility provided by the ANM implementation in LightDock is supported for the ligand (soluble) molecules (not the membrane-embedded proteins as those are considered as one entity together with the beads). This can be activated as: -anm -anm_rec=0 -anm_lig=X, where X indicates the number of non-trivial normal modes to be considered (being 10 the recommended value).”

We would like to thank again the reviewer for the relevant comments and insights.

REVIEWERS' COMMENTS

Reviewer #1 (Remarks to the Author):

Thank you for discussing the comments I raised.